# OSHO-CCA: Orthogonal and Scalable High-Order Canonical Correlation Analysis

## Abstract

Canonical Correlation Analysis (CCA) is a classical technique for learning shared representations from two views of data by maximizing the correlation between the resulting representations. Existing extensions to more than two views either maximize pairwise correlations, sacrificing higher-order structure, or model high-order interactions at the expense of orthogonality and scalability. In this paper, within the scope of linear multiview correlation learning, we propose **OSHO-CCA**, a novel linear method for Orthogonal and Scalable High-Order CCA that jointly addresses all three desiderata: (1) it captures high-order dependencies across views, (2) enforces orthogonality among projected features to ensure decorrelated embeddings, and (3) scales efficiently with the number of views. We further introduce a new evaluation metric for Total Canonical Correlation (TCC) that generalizes traditional two-view CCA metrics to the multiview setting. Experiments on real and synthetic datasets demonstrate that OSHO-CCAoutperforms existing methods in both correlation maximization and downstream classification tasks, while maintaining scalability and orthogonality even in challenging multiview scenarios.

## 1 Introduction

Multiview learning aims to utilize complementary information from multiple data representations, or views, to extract robust features and enhance learning performance. Canonical Correlation Analysis (CCA) (Hotelling, 1992) is a foundational technique in this domain, seeking linear transformations of two sets of variables such that their transformed representations are maximally correlated. CCA has seen widespread applications in diverse fields, including neuroscience (Smith et al., 2015; Miller et al., 2016), genomics and biology (Butler et al., 2018; Witten & Tibshirani, 2009; Stuart et al., 2019), and ecology (Anderson & Willis, 2003). Despite its widespread use, CCA inherently supports only two views, restricting its applicability in scenarios involving multiple heterogeneous data sources. In recent years, multiview methods that can handle more than two views have become increasingly critical for capturing complex relationships across diverse modalities.

To fill this gap, various generalizations of CCA to multiple views have been proposed. These methods may be clustered into two primary groups: earlier approaches that focus on pairwise correlations, and more recent ones that incorporate high-order interactions. The former includes natural extensions of CCA, which focus on maximizing the pairwise correlations between all view pairs (Horst, 1961; Kettenring, 1971; Carroll, 1968), and represent earlier approaches in the development of multiview learning methods. Although these approaches typically have a scalable, closed-form solution, they fail to capture high-order interactions, that arise only when all views are considered jointly. This limits their ability to exploit multiview dependencies fully. For example, in scenarios where information is distributed across three or more modalities, the views may exhibit *pairwise independence*, that is, any two views appear statistically independent, while still jointly encoding meaningful structure. A classic case is the XOR function: any two binary inputs are uncorrelated, but their joint interaction determines the output. This is a simple example that demonstrates how high-order interactions are crucial for fully capturing the underlying data structure.

|  | High-order correlation | Scalability | Orthogonality |
|---|:---:|:---:|:---:|
| SUMCOR | ✗ | ✓ | ✗ |
| MAXVAR | ✗ | ✓ | ✓ |
| TCCA | ✓ | ✗ | ✗ |
| **OSHO-CCA** | ✓ | ✓ | ✓ |

Figure 1: Multiview CCA properties. OSHO-CCA is the only method to have the three required properties of multiview CCA: high-order correlation, scalability and orthogonality.

On the more recent group which incorporate high-order interaction, the main method is Tensor CCA (TCCA) (Luo et al., 2015). TCCA analyzes the high-order covariance tensor formed from the data, rather than relying on pairwise covariance matrices. Despite its strengths, it exhibits two key limitations: the absence of orthogonal representations and limited scalability. Orthogonality plays a crucial role in ensuring that the learned directions within each view are statistically uncorrelated, thereby avoiding redundancy and enabling each component to capture distinct aspects of the shared variation. This principle is fundamental in classical two-view CCA, where the canonical vectors are explicitly constrained to be orthogonal within a view. Without this constraint, the optimization can converge to trivial or degenerate solutions that do not reflect any meaningful correlation structure between the views. Consequently, the learned representations may collapse into overlapping subspaces, reducing their informativeness and not fully achieving the objective of CCA. Second, computing the full covariance tensor is computationally inefficient and memory-intensive, as its size scales with $\prod_{v=1}^{V} D_v$, where $V$ is the number of views and $D_i$ is the dimension of the $v$-th view. Consequently, the complexity of TCCA grows exponentially with the number of views, preventing its use in practice.

To overcome these limitations, within the linear multiview correlation learning regime, we propose OSHO-CCA, which succeeds in achieving all three properties together (see Fig. 1): it seeks to find canonical variables that maximize the **high-order correlation** across all views; it is a **scalable** approach with respect to both the dimensionality and the number of views; and it enforces an **orthogonality** constraint on the canonical variables to ensure their decorrelation. The key insight that enables us to achieve orthogonality is that, via an architectural design, we managed to convert a constrained optimization into an unconstrained one. To the best of our knowledge, this is the first method to achieve all three properties together. Moreover, inspired by the high-order correlation objective of TCCA, we describe a total canonical correlation (TCC) evaluation metric that takes into account the orthogonality of the canonical variables. We then demonstrate the effectiveness of leveraging high-order interactions over pairwise interactions through a synthetic experiment. We further present experimental results on multiview datasets which demonstrate that OSHO-CCA consistently performs on par with and often better than other SOTA approaches for linear multiview CCA in terms of both classification accuracy and TCC, while ensuring the orthogonality constraint.

## 2 Related Work

In this section, we describe the main approaches to multiview CCA corresponding to the two previously introduced categories: (1) pairwise correlation-based methods and (2) high-order interaction-based methods. We also discuss the current evaluation methods to asses multiview CCA.

## 2.1 Multiview CCA Methods

Among the pairwise correlation-based approaches, two of the earliest and most well-known methods are SUMCOR-GCCA (Horst, 1961; Kettenring, 1971) and MAXVAR-GCCA (Carroll, 1968). SUMCOR-GCCA maximizes the sum of all pairwise correlations among the views, providing a direct extension of classical CCA. In contrast, MAXVAR-GCCA aims to uncover a shared latent structure by finding a common representation that best aligns with each view individually. LS-CCA (Vía et al., 2007) further develops this line by minimizing the distances between projected views in a least-squares sense, and is shown to be equivalent to MAXVAR-GCCA under specific formulations.

To unify these multivariate approaches, Regularized Generalized Canonical Correlation Analysis (RGCCA) was proposed as a flexible framework that deals with multiview vector-valued data (Tenenhaus & Tenenhaus, 2011; Tenenhaus et al., 2017). However it does not assume or account for higher-order tensor structures, treating all views as vector inputs. To incorporate these structures, Min et al. (2019) introduced a two-view approach that treats data as explicitly tensorial and imposes a rank-$R$ CP decomposition on the canonical tensors, a concept later generalized to the multiview setting by Girka et al. (2024), which extended it by also imposing an orthogonal rank-$R$ CP decomposition. However, similar to RGCCA and other methods like SUMCOR and MAXVAR, these tensor-based generalizations still operate by optimizing criteria based on the aggregation of pairwise correlations or covariances, rather than explicitly modeling the simultaneous higher-order dependencies across all views jointly.

To move beyond the limitations of pairwise correlations, high-order methods such as TCCA (Luo et al., 2015) have been proposed. TCCA explicitly models joint interactions among all views by maximizing a high-order correlation objective defined over a covariance tensor. TCCA directly maximizes the multi-view correlation but comes with significant computational challenges: the construction and decomposition of the covariance tensor can be memory- and time-intensive, scaling exponentially with the number of views. Moreover, TCCAs solutions are not guaranteed to be orthogonal, potentially leading to redundancy among the learned representations.

In summary, existing linear methods for multiview learning based on CCA either prioritize computational efficiency at the cost of modeling only pairwise correlations or attempt to capture high-order correlations but suffer from scalability and non-orthogonality issues. These limitations motivate the need for new approaches that can efficiently model high-order interactions across multiple views while ensuring orthogonality and maintaining tractable computational complexity. Our proposed method, OSHO-CCA, aims to address these challenges, offering a scalable, principled solution that better captures the underlying high-order multiview structure.

## 2.2 Evaluation for Multiview CCA

While most existing evaluations of multiview CCA methods focus on downstream classification accuracy (Luo et al., 2015; Wong et al., 2021), we argue that this metric alone is not sufficient for assessing the quality of the learned representations in this setting. Accuracy reflects performance on a specific supervised task and may fail to capture the underlying correlation structure across the different views. In classical two-view CCA, the most common evaluation metric is the TCC (Andrew et al., 2013; Chapman et al., 2023b; Michaeli et al., 2016), which quantifies the overall correlation between the projected views while the features in each view are decorrelated. However, in the case of high-order multiview CCA, it is not straightforward to extend this notion, and there is no standard way to measure the TCC across more than two views. Inspired by the formulation of TCC introduced by Luo et al. (2015), we introduce an additional evaluation metric that generalizes the TCC to the multiview case. This metric provides a more intrinsic and task-independent measure of how well the views are aligned and decorrelated in the projected space.

# 3 Preliminaries

**Notations.** Let $\mathbf{X}_1, \mathbf{X}_2, \ldots, \mathbf{X}_V$ be a collection of $V$ matrices, where each $\mathbf{X}_v \in \mathbb{R}^{D_v \times N}$. We denote the $i$-th row of $\mathbf{X}_v$ by $\mathbf{x}_{v,i}{}^\top \in \mathbb{R}^N$ and the $j$-th column by $\mathbf{x}_{v,\cdot j} \in \mathbb{R}^{D_i}$. Throughout, we also adopt

standard tensor notations as in De Lathauwer et al. (2000). We denote the outer product of $V$ vectors $\mathbf{u}_1 \in \mathbb{R}^{d_1}, \mathbf{u}_2 \in \mathbb{R}^{d_2}, \ldots, \mathbf{u}_V \in \mathbb{R}^{d_V}$ as a $V$-way tensor $\mathcal{A} = \mathbf{u}_1 \circ \mathbf{u}_2 \circ \cdots \circ \mathbf{u}_V \in \mathbb{R}^{d_1 \times d_2 \times \cdots \times d_V}$ with entries defined elementwise as:

$$\mathcal{A}(i_1, i_2, \ldots, i_V) = \prod_{v=1}^{V} \mathbf{u}_v(i_v).$$

The $p$-mode product of a tensor $\mathcal{A} \in \mathbb{R}^{I_1 \times \cdots \times I_p \times \cdots \times I_m}$ with a matrix $\mathbf{U} \in \mathbb{R}^{J \times I_p}$ is denoted by $\mathcal{B} = \mathcal{A} \times_p \mathbf{U}$, resulting in a tensor $\mathcal{B} \in \mathbb{R}^{I_1 \times \cdots \times I_{p-1} \times J \times I_{p+1} \times \cdots \times I_m}$ with entries

$$\mathcal{B}(i_1, \ldots, i_{p-1}, j, i_{p+1}, \ldots, i_m) = \sum_{i_p=1}^{I_p} \mathcal{A}(i_1, \ldots, i_p, \ldots, i_m) \cdot \mathbf{U}(j, i_p).$$

We also denote sequential $p$-mode products by: $\mathcal{A} \times_1 \mathbf{U}_1 \times_2 \mathbf{U}_2 \cdots \times_m \mathbf{U}_m$, where each $\mathbf{U}_p \in \mathbb{R}^{J_p \times I_p}$. At last, we denote $\odot$ to be the element-wise product of vectors.

### 3.1 Tensor Canonical Correlation Analysis

Given $V$ centered views of data, where the $v$-th view is represented by a data matrix $\mathbf{X}_v \in \mathbb{R}^{D_v \times N}$, with $D_v$ denoting the dimension and $N$ the number of samples, the goal of TCCA is to learn linear projection matrices $\mathbf{W}_v \in \mathbb{R}^{D_v \times k}$ that maximize the high-order canonical correlation projected variables $\mathbf{Y}_v = \mathbf{W}_v^\top \mathbf{X}_v$.

Formally, TCCA aims to optimize the high-order canonical correlation, which is defined as:

$$\max_{\mathbf{W}_1, \ldots, \mathbf{W}_V} \rho(\mathbf{Y}_1, \ldots, \mathbf{Y}_V) \tag{1}$$
$$\text{s.t.} \quad \mathbf{Y}_v \mathbf{Y}_v^\top = I_k \quad \forall v = 1, \ldots, V$$

where the high-order correlation (not canonical) $\rho$ is defined as:

$$\rho(\mathbf{Y}_1, \ldots, \mathbf{Y}_V) \triangleq \sum_{i=1}^{k} \left( \mathbf{y}_{1,i\cdot} \odot \mathbf{y}_{2,i\cdot} \odot \cdots \odot \mathbf{y}_{V,i\cdot} \right) \mathbf{e} \tag{2}$$

and $\mathbf{e} \in \mathbb{R}^N$ is an all-one column vector. For intuition, in the two-view case, equation 1 reduces to the sum of Pearson correlation coefficients across the $k$ dimensions. For further refinement, a construction of the high-order covariance tensor is introduced:

$$\mathcal{C} = \frac{1}{N} \sum_{n=1}^{N} \mathbf{x}_{1,\cdot n} \circ \mathbf{x}_{2,\cdot n} \circ \cdots \circ \mathbf{x}_{V,\cdot n} \in \mathbb{R}^{D_1 \times D_2 \times \cdots \times D_V}.$$

Which leads to a Rayleigh quotient-like optimization problem equivalent to equation 1:

$$\max_{\mathbf{W}_1, \ldots, \mathbf{W}_V} \sum_{j=1}^{k} \mathcal{C} \times_1 \mathbf{w}_{1,\cdot j}^\top \times_2 \mathbf{w}_{2,\cdot j}^\top \cdots \times_V \mathbf{w}_{V,\cdot j}^\top \tag{3}$$
$$\text{s.t.} \quad \mathbf{Y}_v \mathbf{Y}_v^\top = I_k \quad \forall v = 1, \ldots, V.$$

Define for each view $v$, the covariance matrix $\mathbf{\Sigma}_v \in \mathbb{R}^{D_v \times D_v}$, $\mathbf{Q}_v := \mathbf{\Sigma}_v^{\frac{1}{2}} W_v$, and the correlation tensor $\mathcal{M} = \mathcal{C} \times_1 \mathbf{\Sigma}_1^{-\frac{1}{2}} \times_2 \mathbf{\Sigma}_2^{-\frac{1}{2}} \cdots \times_V \mathbf{\Sigma}_V^{-\frac{1}{2}}$. Then, equation 3 can be reformulated to:

$$\max_{\mathbf{Q}_1, \ldots, \mathbf{Q}_V} \sum_{j=1}^{k} \mathcal{M} \times_1 \mathbf{q}_{1,\cdot j}^\top \times_2 \mathbf{q}_{2,\cdot j}^\top \cdots \times_V \mathbf{q}_{V,\cdot j}^\top \tag{4}$$
$$\text{s.t.} \quad \mathbf{Q}_v^\top \mathbf{Q}_v = I_k \quad \forall v = 1, \ldots, V.$$

In the two-view case, this optimization problem reduces to a Rayleigh quotient defined over the correlation matrix. In practice, TCCA performs a rank-$k$ tensor approximation on $\mathcal{M}$ via Alternating Least Squares (ALS), to extract the canonical directions. However, two critical limitations arise: (1) **Lack of orthogonality** While TCCA's first problem formulation comes with an orthogonal representations (equation 1) constraint, the authors admit that it fails to achieve it, as the resulting projection matrices $\mathbf{Q}_1, \ldots, \mathbf{Q}_V$ are not orthogonal. As shown by De Lathauwer et al. (2000), the best rank-$k$ approximation of a tensor does not, in general, produce orthogonal components. This stands in contrast to the matrix case, where the top $k$ singular vectors are orthogonal by construction and yield the optimal rank-$k$ approximation. (2) **Computational complexity** Constructing and decomposing $\mathcal{M}$ becomes infeasible for high-dimensional data or a large number of views, due to its size of $\prod_{v=1}^{V} D_v$.

These limitations motivate the development of more efficient and principled methods, such as our proposed OSHO-CCA, that retain the strengths of TCCA namely, the high-order correlation modeling while enforcing orthogonality and improving scalability.

## 4 OSHO-CCA

### 4.1 Rationale

In this section, we introduce the proposed method, OSHO-CCA, which overcomes the two key limitations of TCCA, namely the lack of orthogonality and poor scalability, while preserving the benefits of modeling high order interactions. Instead of explicitly constructing the full correlation tensor, OSHO-CCA directly optimizes the objective in equation 1 via gradient descent. Crucially, we turn the original constrained optimization problem into an equivalent unconstrained one through an architectural design that enforces orthogonality by construction, inspired by Shaham et al. (2018). Instead of adding orthogonality as a penalty term in the loss, which may not strictly enforce the constraint, our design ensures that optimization remains confined to the space of orthogonal solutions throughout training. This allows us to overcome the core obstacle that prevents TCCA from achieving orthogonal projections. Notably, in the special case of two views, OSHO-CCA reduces to classical CCA, preserving consistency with the standard formulation in the bivariate setting.

### 4.2 Method Formulation

Following the notations from TCCA, we denote the covariance matrix for each view $v$ as $\boldsymbol{\Sigma}_v \in \mathbb{R}^{D_v \times D_v}$, and introduce the transformed variable $\mathbf{Q}_v := \boldsymbol{\Sigma}_v^{\frac{1}{2}} \mathbf{W}_v$, which implies that $\mathbf{Y}_v = \mathbf{W}_v^\top \mathbf{X}_v = \mathbf{Q}_v^\top \boldsymbol{\Sigma}_v^{-\frac{1}{2}} \mathbf{X}_v$. Using this, the objective in equation 1 can be reformulated as:

$$\max_{\mathbf{Q}_1, \ldots, \mathbf{Q}_V} \rho(\mathbf{Y}_1, \ldots, \mathbf{Y}_V) \tag{5}$$
$$\text{s.t.} \quad \mathbf{Q}_v \mathbf{Q}_v^\top = I_k \quad \forall v = 1, \ldots, V$$

Here, the projection matrices $\mathbf{Q}_v$ are constrained to be orthogonal. To transform the above constrained optimization problem into an unconstrained one, we express $\mathbf{Q}_v$ as the orthonormal component of the QR decomposition of some learnable matrices $\mathbf{U}_v$, i.e., $\mathbf{U}_v = \mathbf{Q}_v \mathbf{R}_v$. Substituting this into the objective leads to the final *uncostrained* optimization problem:

$$\max_{\mathbf{U}_1, \ldots, \mathbf{U}_V} \rho(\mathbf{Y}_1, \ldots, \mathbf{Y}_V). \tag{6}$$

The key idea is to optimize over auxiliary matrices $\mathbf{U}_v$ and extract the desired orthogonal matrices $\mathbf{Q}_v$ through their QR decomposition. Since the QR decomposition is differentiable, this approach enables efficient optimization via gradient descent (Shaham et al., 2018). It is important to emphasize that we utilize the QR decomposition primarily as a functional tool to enforce the required orthogonality constraints in a differentiable manner, rather than as a core theoretical focus of the method. The full procedure is detailed in Alg. 1.

---

**Algorithm 1** OSHO-CCA

---

**Input:** Data views $\mathbf{X}_1, \mathbf{X}_2, \ldots, \mathbf{X}_V$; number of iterations $T$
**Output:** Projection matrices $\mathbf{W}_1, \ldots, \mathbf{W}_V$
 1: Randomly initialize $\mathbf{U}_v \in \mathbb{R}^{D_v \times k}$ for $v = 1, \ldots, V$
 2: Compute whitening matrices $\boldsymbol{\Sigma}_1^{-\frac{1}{2}}, \ldots, \boldsymbol{\Sigma}_V^{-\frac{1}{2}}$
 3: **for** iteration $t = 1$ to $T$ **do**
 4:    **for** each view $v = 1$ to $V$ **do**
 5:       Perform QR decomposition: $\mathbf{U}_v = \mathbf{Q}_v \mathbf{R}_v$
 6:       Update: $\mathbf{W}_v^\top = \mathbf{Q}_v^\top \boldsymbol{\Sigma}_v^{-\frac{1}{2}}$
 7:       Compute projected features: $\mathbf{Y}_v = \mathbf{W}_v^\top \mathbf{X}_v$
 8:    **end for**
 9:    Compute the loss $\mathcal{L} = -\rho(\mathbf{Y}_1, \ldots, \mathbf{Y}_V)$
10:    Update $\mathbf{U}_1, \ldots, \mathbf{U}_V$ using gradient descent
11: **end for**
12: **return** $\mathbf{W}_1, \ldots, \mathbf{W}_V$

---

### 4.3 Total Canonical Correlation Evaluation for High-Order Methods

As discussed earlier, the dominant evaluation strategy for high-order multiview CCA methods relies on downstream linear classification accuracy. However, accuracy alone does not reflect the degree of correlation or alignment across the views. Drawing inspiration from the classical two-view CCA setting where TCC serves as a key metric, we introduce an analogous evaluation measure for the multiview case. This metric is designed to jointly assess inter-view correlation and intra-view decorrelation of the learned representations.

While equation 2 provides a formal definition of the high-order total correlation, it does not enforce orthogonality among the representations, and thus cannot be considered canonical. To ensure orthogonality is considered, we first apply the Gram-Schmidt process to each view to orthonormalize its features. We then evaluate the high-order TCC using equation 2, thereby obtaining a metric that faithfully captures both the shared structure across views and the orthogonality within each view. The formal definition for TCC is:

$$\text{TCC}(\mathbf{Y}_1, \ldots, \mathbf{Y}_V) = \rho(\tilde{\mathbf{Y}}_1, \ldots, \tilde{\mathbf{Y}}_V). \tag{7}$$

Where $\tilde{\mathbf{Y}}_v$ denotes the matrix obtained by applying the Gram-Schmidt process to the feature vectors of $\mathbf{Y}_v$.

## 5 Experiments

### 5.1 Evaluation Metrics

#### 5.1.1 Total Canonical Correlation Evaluation

We used the proposed TCC evaluation defined in equation 7. We observed that the TCC decreases significantly as the number of views increases. To address this, we applied a simple normalization (see Appendix).

#### 5.1.2 Classification Accuracy

Following previous works (Luo et al., 2015; Wong et al., 2021), we use Classification Accuracy (ACC) to assess the practical utility of the learned canonical variables.

For classification, the embeddings from all $V$ views of each sample are concatenated into a single feature vector. This combined vector then serves as input for a linear classifier. Specifically, we employed a Linear Support Vector Classifier (SVC). The regularization term $C$ for the SVC was chosen as the best performing value on the validation set, selected from the interval [0.01,0.1,0.5,1]. We train the classifier on the training data and report its accuracy on the test set, providing an objective measure of the discriminative power of each method's learned representations.

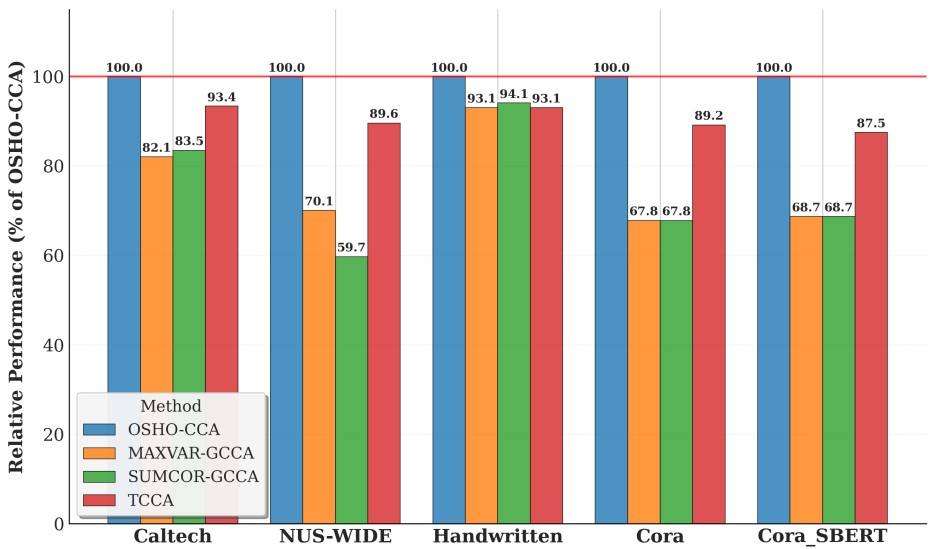

Figure 2: TCC of baseline methods as a percentage of OSHO-CCA's TCC

## 5.2 Datasets

To evaluate our proposed method, we utilized four standard benchmarks multi-view datasets: Caltech-20 (Li et al., 2022), Handwritten (Duin, 1998), NUS-WIDE (wang xiao yan & Lu, 2023), Cora (Fang et al., 2023), and a variant of Cora, which will be denoted as Cora_SBERT. The feature extraction procedures for the different datasets follow those described in Li et al. (2015). A detailed description of each dataset is provided in the Appendix.

## 5.3 Baselines

For evaluation, we compared OSHO-CCA against several SOTA approaches for multiview CCA. All beselines are implemented using the CCA-Zoo package (Chapman et al., 2023a). These baselines represent different strategies for extending CCA to multiple views. In particular, we compare to **SUMCOR-GCCA** (Horst, 1961; Kettenring, 1971), which maximizes the sum of all pairwise correlations,**MAXVAR-GCCA** (Carroll, 1968), which finds a common latent representation that best correlates to all views, and **TCCA** (Luo et al., 2015), which aims at capturing high-order interactions. Regarding TCCA, to ensure applicability across all datasets used in the experiments, we first apply PCA views, reducing their dimensionality to ensure TCCA remains computationally feasible.

## 5.4 Experimental Setup

For each dataset, we followed a consistent data splitting protocol. The data was partitioned into a training set (70%), a validation set (15%), and a test set (15%). To ensure the robustness and reliability of our results, we performed 5 independent runs for each experiment, using different random splits of the data. The exact same splits were used across all baseline methods to guarantee a fair and direct comparison. In all experiments, we set the output dimensions $k = 10$, except for the Handwritten dataset, where the smallest feature dimension is 6; thus, we set $k = 6$. Prior to training, all feature views for each dataset were standard normalized. For OSHO-CCA, we applied PCA to each view before training to improve optimization stability and overall performance. We also tested PCA on the baseline methods. However, it did not improve the performance on their pairwise correlation.

Table 1: Comparison of model results on different datasets in Terms of classification accuracy (ACC)

| Dataset | Model | ACC |
|---|---|---|
| Caltech-20 | MAXVAR-GCCA | $0.764 \pm 0.184$ |
| | SUMCOR-GCCA | $0.857 \pm 0.012$ |
| | TCCA | $0.868 \pm 0.011$ |
| | OSHO-CCA | $\mathbf{0.887 \pm 0.009}$ |
| NUS-WIDE | MAXVAR-GCCA | $0.325 \pm 0.008$ |
| | SUMCOR-GCCA | $\mathbf{0.395 \pm 0.008}$ |
| | TCCA | $0.368 \pm 0.006$ |
| | OSHO-CCA | $0.392 \pm 0.005$ |
| Handwritten | MAXVAR-GCCA | $0.804 \pm 0.015$ |
| | SUMCOR-GCCA | $0.920 \pm 0.015$ |
| | TCCA | $0.930 \pm 0.012$ |
| | OSHO-CCA | $\mathbf{0.944 \pm 0.019}$ |
| Cora | MAXVAR-GCCA | $0.346 \pm 0.013$ |
| | SUMCOR-GCCA | $\mathbf{0.427 \pm 0.012}$ |
| | TCCA | $0.389 \pm 0.032$ |
| | OSHO-CCA | $0.414 \pm 0.053$ |

## 5.5 Results

Fig. 2 presents the results of the TCC experiment, showing TCC of the baseline methods as a percentage of OSHO-CCAs TCC. As can be seen, OSHO-CCA consistently attains the highest TCC across all datasets. This suggests that it is particularly effective in aligning multiple views into a common subspace, capturing shared structure more robustly than the baselines.

In terms of classification accuracy (Table 1), OSHO-CCA performs competitively across all datasets. While it achieves the best performance in half of the datasets, it remains close to the strongest baseline even when not leading. This indicates that the embeddings produced by OSHO-CCA not only capture correlation well but also retain discriminative power for downstream tasks.

Together, the results demonstrate that OSHO-CCA is a strong and robust alternative for multiview representation learning.

## 5.6 High-Order Correlation Experiment

To motivate the use of high-order correlation methods, we construct a synthetic dataset in which all pairs of views are (approximately) uncorrelated, yet meaningful dependencies emerge when considering their joint interaction. This scenario highlights the limitations of pairwise correlation methods such as MAXVAR-GCCA and SUMCOR-GCCA, and demonstrates the necessity of using high-order methods like TCCA and OSHO-CCA.

Each sample is generated by independently drawing two latent binary vectors $\mathbf{x}, \mathbf{y} \in \{-1, +1\}^2$, sampled uniformly from the set of four binary corners. A third latent vector $\mathbf{z}$ is defined as the elementwise product: $\mathbf{z} = \mathbf{x} \odot \mathbf{y}$. To obscure trivial alignments and simulate real-world noise, we apply distinct random orthogonal transformations $\mathbf{Q}_1$ and $\mathbf{Q}_2$ to $\mathbf{x}$ and $\mathbf{y}$, respectively, and add independent Gaussian noise to all three views:

$$\mathbf{x}' = \mathbf{Q}_1 \mathbf{x} + \boldsymbol{\epsilon}_1, \qquad \mathbf{y}' = \mathbf{Q}_2 \mathbf{y} + \boldsymbol{\epsilon}_2,$$

$$\mathbf{z}' = \mathbf{x} \odot \mathbf{y} + \boldsymbol{\epsilon}_3, \qquad \boldsymbol{\epsilon}_v \sim \mathcal{N}\left(0, \left(\frac{1}{10}\right)^2 \mathbf{I}\right).$$

Since $\mathbf{x}$ and $\mathbf{y}$ are independently sampled, and $\mathbf{z}$ is deterministically defined from both, all pairwise correlations between the transformed views $(\mathbf{x}', \mathbf{y}')$, $(\mathbf{x}', \mathbf{z}')$, and $(\mathbf{y}', \mathbf{z}')$ are close to zero in expectation. As a result, the pairwise multiview methods MAXVAR-GCCA and SUMCOR-GCCA are unable to recover meaningful

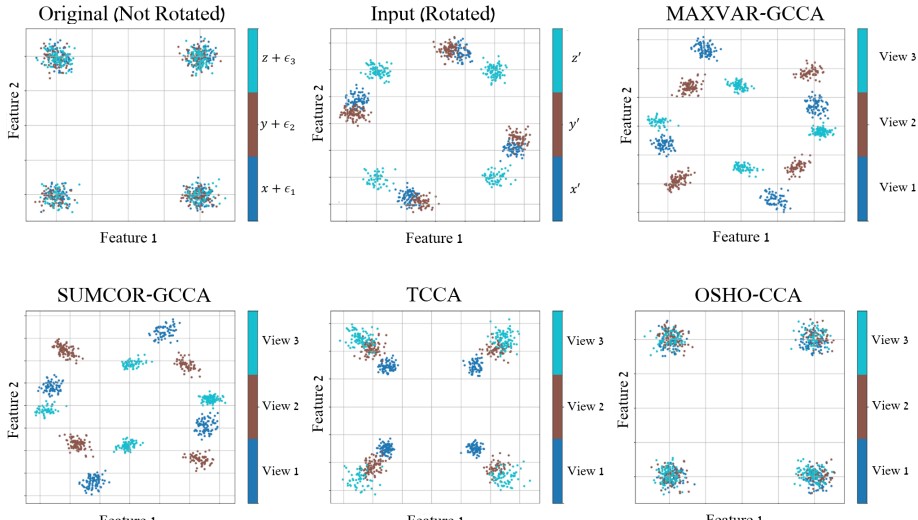

Figure 3: Pairwise correlation methods fail to achieve aligned representations like high-order ones. The plot shows the two coordinates of each view, with colors representing different views. The top-left plot depicts the original (unrotated) views, while the top-middle plot shows their rotated version, which serves as the input to all methods. The goal is to recover an aligned representation that matches the original views. MAXVAR-GCCA and SUMCOR-GCCA fail to achieve this, resulting in near-zero pairwise correlations due to their reliance on pairwise dependencies. In contrast, OSHO-CCA and TCCA successfully recover the correct joint alignment, capturing the shared high-order interactions across all views.

representations. In contrast, OSHO-CCA which exploits high-order interactions across all three views, can successfully recover the latent signal.

Fig. 3 illustrates the high-order experiment, where every color represents a different view, showing the embeddings produced by each method. As expected, both MAXVAR-GCCA and SUMCOR-GCCA fail to recover meaningful structure, yielding near-zero pairwise correlations due to the statistical independence between all pairs of views. In contrast, OSHO-CCA and TCCA successfully uncover the correct joint alignment by recovering the appropriate rotations of **x** and **y**, effectively capturing the high-order interaction shared across all three views.

## 5.7 Orthogonality Experiment

As previously discussed, the TCCA solution involves a rank-$k$ approximation of the correlation tensor. However, as shown by De Lathauwer et al. (2000), the best rank-$k$ approximation and orthogonality cannot be achieved simultaneously. Consequently, the TCCA solution is inherently non-orthogonal, despite the original formulation aiming to recover an orthogonal solution analogous to the two-view CCA setting. To illustrate this limitation, we conducted an experiment varying the number of views from two to six. For each configuration, we trained both TCCA and OSHO-CCA on the same views from the MFEAT dataset and plotted the covariance matrices of the transformed FOU view. The results, shown in Fig. 4, highlight this effect: the top row shows the transformed views from TCCA, while the bottom row corresponds to OSHO-CCA. As the number of views increases, TCCA produces increasingly non-orthogonal representations, whereas OSHO-CCA consistently maintains orthogonality.

## 5.8 Scalability Experiment

In this section, we evaluate the running time of TCCA and OSHO-CCA as the number of views increases. For that, we construct a synthetic multiview dataset based on a shared latent representation. We begin by generating $N$ samples, each associated with a discrete class label $y_i \in \{1, \ldots, C\}$. Each class $c$ is represented

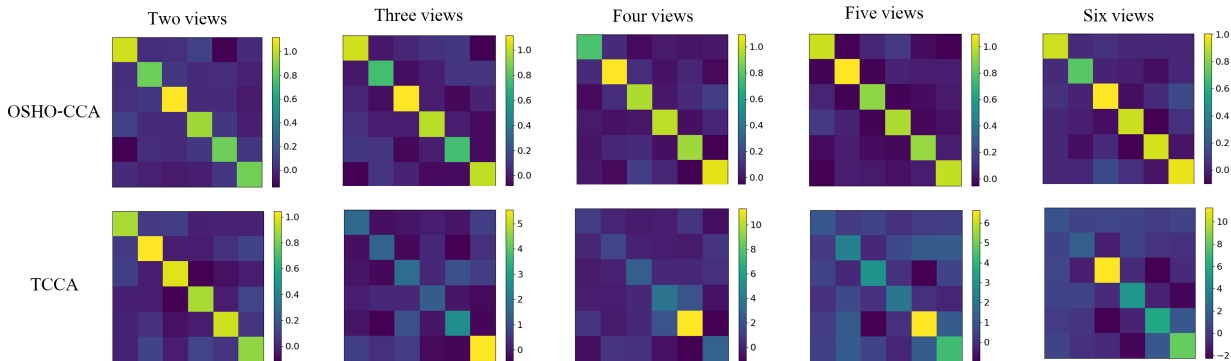

Figure 4: OSHO-CCA vs. TCCA in terms of orthogonality. This figure shows the covariance matrices of the transformed features from the FOU view of the Handwritten test set, used to assess orthogonality. The top row shows results from OSHO-CCA, and the bottom row from TCCA, across varying numbers of views. While both methods maintain orthogonality with two views, TCCA increasingly fails to preserve orthogonal representations as the number of views grows, whereas OSHO-CCA consistently succeeds.

by a mean vector $\boldsymbol{\mu}_c \in \mathbb{R}^{d_z}$ in a latent space of dimension $d_z$. For each sample $i$, we draw a latent vector $\mathbf{z}_i \in \mathbb{R}^{d_z}$ from a Gaussian distribution centered at the class mean $\mathbf{z}_i \sim \mathcal{N}(\boldsymbol{\mu}_{y_i}, \sigma_z^2\mathbf{I})$, where $\sigma_z^2$ controls intra-class variability. To simulate $m$ distinct views, we construct random linear projection matrices $\mathbf{P}_1, \ldots, \mathbf{P}_V$, where $\mathbf{P}_v \in \mathbb{R}^{D_v \times d_z}$. Each view $v$ is generated by projecting the latent vector and adding Gaussian noise:

$$\mathbf{x}_{v,\cdot n} = \mathbf{P}_v\mathbf{z}_n + \boldsymbol{\epsilon}_{v,\cdot n}, \quad \boldsymbol{\epsilon}_{v,\cdot n} \sim \mathcal{N}(0, \sigma^2\mathbf{I}),$$

where $\sigma^2$ controls the noise level. This process yields $V$ views $\mathbf{X}_1, \ldots, \mathbf{X}_V$, with $\mathbf{X}_v \in \mathbb{R}^{D_v \times N}$, each offering a distinct view of the same latent structure. The shared signal $\mathbf{z}_n$ induces cross-view correlation, while the projections and noise introduce view-specific diversity. In this experiment, we varied the number of views $V$ from two to six, using a total of $N = 2000$ samples, split into 1400 for training and 300 each for validation and testing. Each view was set to have dimension $D_v = 20$, with a shared latent space of dimension $d_z = 10$. Both methods were projected onto this latent space for a fair comparison. Fig. 5 presents the results, showing both the runtime (right) and the relative improvement in TCC (left) of OSHO-CCA over TCCA as the number of views increases. As shown in the right plot, the runtime of both methods is comparable for a small number of views. However, TCCA's runtime grows exponentially with the number of views, and we were unable to run TCCA for seven views due to prohibitive memory and computational costs associated with constructing and decomposing a seventh-order tensor. The left plot highlights the increasing advantage of OSHO-CCA over TCCA in terms of TCC, attributed to TCCAs inability to maintain orthogonality as the number of views grows.

## 5.9 Ablation Study

To evaluate the necessity of the orthogonality constraint, we conducted an ablation study by removing the QR decomposition step from OSHO-CCA. Without this structural constraint, the model cannot explicitly guarantee decorrelated embeddings. As detailed in Table 2, the absence of the orthogonal decomposition significantly degrades downstream classification accuracy. These findings empirically validate that enforcing orthogonality is critical.

## 6 Conclusion and Future Work

In this work, we introduced OSHO-CCA, a scalable and orthogonal approach to high-order multiview CCA. Unlike prior methods, it captures joint correlations across all views while maintaining orthogonality and

Table 2: Ablation study on the effect of the QR decomposition on classification accuracy (ACC).

| Dataset | w/o QR (ACC) | OSHO-CCA (ACC) |
|---|---|---|
| Handwritten | $0.876 \pm 0.079$ | $\mathbf{0.944 \pm 0.019}$ |
| Caltech-20 | $0.586 \pm 0.03$ | $\mathbf{0.887 \pm 0.009}$ |

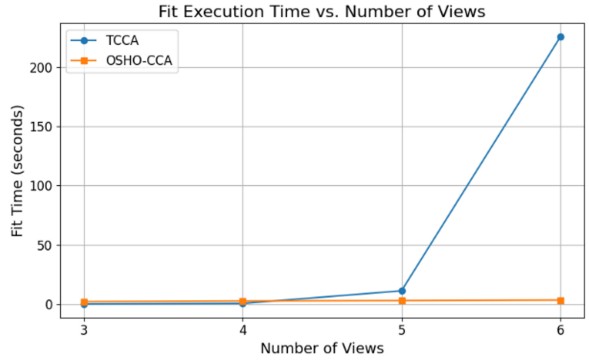 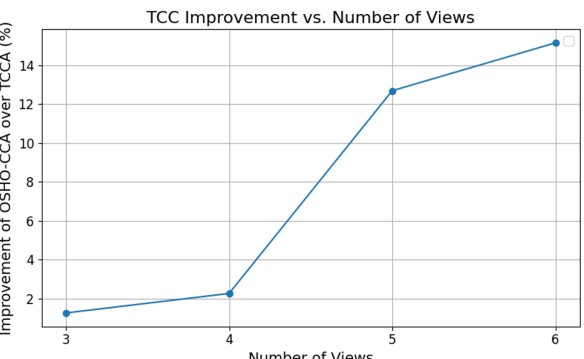

Figure 5: Scalability experiment. The right plot shows the relative improvement of OSHO-CCA over TCCA in terms of TCC as the number of views increases, while the left plot illustrates the runtime of both methods. OSHO-CCA scales efficiently with the number of views, whereas TCCA becomes computationally infeasible beyond six views due to the exponential growth in tensor size.

avoiding the computational overhead of tensor decomposition. Experiments show that OSHO-CCA achieves superior alignment and competitive downstream performance across diverse datasets.

**Future Work** A natural next step is to extend OSHO-CCA to a nonlinear variant, potentially using deep neural networks to achieve improved alignment. This would allow modeling complex relationships while retaining high-order alignment and orthogonality. We also aim to explore mini-batch training and end-to-end supervised variants to improve scalability and task-specific performance.

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

## A Two-View Evaluation of OSHO-CCA

To further validate the effectiveness of OSHO-CCA, we conducted experiments in the classical two-view setting, comparing its performance to CCA. This setup highlights how OSHO-CCA naturally extends CCA when applied to two views.

We used the Handwritten and Caltech-20 datasets, selecting the FAC and FOU views for the former, and the GAB and WM views for the latter. The results, shown in Table 3, demonstrate that OSHO-CCA performs on par with CCA in terms of TCC, supporting its applicability and competitiveness in the two-view regime.

Table 3: Comparison of Model Results on Different Datasets in Terms of TCC.

| Dataset | Model | $TCC$ |
|---------|-------|-------|
| Caltech | CCA | $4.74 \pm 0.30$ |
|         | OSHO-CCA | $4.68 \pm 0.31$ |
| Handwritten | CCA | $5.03 \pm 0.04$ |
|         | OSHO-CCA | $5.02 \pm 0.04$ |

## B Normalized TCC: Definition and Full Results

As described in the Experiments section, we applied a normalization for the TCC value which depends on the number of views $V$, the number of samples $N$, and the dimension size $k$. The normalized TCC defined as:

$$\text{TCC}_{norm} = \sqrt[V]{\frac{\text{TCC}(\mathbf{Y}_1, \ldots, \mathbf{Y}_V)}{kN^{\frac{V}{2}}}} \tag{8}$$

Table 4 presents the absolute results corresponding to Fig. 4 in the main paper, using the normalized TCC.

## C Datasets Details

Each dataset was represented using multiple feature views to capture diverse and complementary information:

**Caltech-20** The Caltech-20 dataset is a subset of the well-known Caltech-101 Li et al. (2022) image collection, selected to comprise 20 distinct object categories. For our experiments, this dataset includes approximately 2,386 images with varying numbers of images per category. Each image is represented using six types of visual features: 48-dimensional Gabor features (GAB), 40-dimensional wavelet moments (WM), 254-dimensional CENTRIST descriptors, 1,984-dimensional Histogram of Oriented Gradients (HOG), 512-dimensional GIST features, and 928-dimensional Local Binary Patterns (LBP).

Table 4: Comparison of Model Results on Different Datasets in Terms of TCC.

| Dataset | Model | TCC$_{norm}$ |
|---|---|---|
| Caltech | MAXVAR-GCCA | $0.234 \pm 0.026$ |
| | SUMCOR-GCCA | $0.238 \pm 0.032$ |
| | TCCA | $0.266 \pm 0.029$ |
| | OSHO-CCA | $\mathbf{0.285 \pm 0.033}$ |
| NUS-WIDE | MAXVAR-GCCA | $0.054 \pm 0.010$ |
| | SUMCOR-GCCA | $0.046 \pm 0.004$ |
| | TCCA | $0.069 \pm 0.003$ |
| | OSHO-CCA | $\mathbf{0.077 \pm 0.002}$ |
| Handwritten | MAXVAR-GCCA | $0.188 \pm 0.002$ |
| | SUMCOR-GCCA | $0.190 \pm 0.002$ |
| | TCCA | $0.188 \pm 0.020$ |
| | OSHO-CCA | $\mathbf{0.202 \pm 0.015}$ |
| Cora | MAXVAR-GCCA | $0.331 \pm 0.003$ |
| | SUMCOR-GCCA | $0.331 \pm 0.003$ |
| | TCCA | $0.435 \pm 0.054$ |
| | OSHO-CCA | $\mathbf{0.488 \pm 0.058}$ |

**Handwritten**  The Handwritten dataset Duin (1998), also known as the "Multiple Features" dataset from the UCI Repository, consists of 2,000 samples of handwritten digits from 0 to 9, corresponding to 10 classes. Each sample is described by six different feature sets: 76-dimensional Fourier coefficients (FOU), 216-dimensional profile correlations (FAC), 64-dimensional Karhunen-Loève coefficients (KAR), 240-dimensional pixel averages (PIX), 47-dimensional Zernike moments (ZER), and 6-dimensional morphological features (MOR).

**NUS-WIDE**  The NUS-WIDE dataset wang xiao yan & Lu (2023) is a subset of the larger NUS-WIDE database. The version used in our experiments consists of 26,315 images annotated with 31 semantic concepts (classes). Each image is described by five feature views: 65-dimensional color histograms (CH), 226-dimensional color moments (CM), 145-dimensional color correlograms (CORR), 74-dimensional edge direction histograms (EDH), and 129-dimensional wavelet textures (WT).

**Cora_SBERT**  The Cora_SBERT variant replaces the 1,433-dimensional sparse bag-of-words representation of the original Cora dataset with dense embeddings generated by Sentence-BERT (SBERT). While the textual feature view is upgraded to capture deeper semantic context, the structural views, including the 2,708-dimensional inbound, outbound, and combined citation indicators, remain identical to the standard Cora dataset. This setup enables a direct comparison between keyword-based and transformer-based feature representations within the multiview framework. The data has taken from Wang et al. (2025)

# D   Technical Details

For all methods, we regularize each view-specific covariance matrix by adding a small ridge term $\widetilde{\Sigma}_v = \Sigma_v + \epsilon I$, where $I$ is the identity matrix and we set $\epsilon = 10^{-6}$ in all experiments to ensure positive definiteness and numerical stability. Regarding OSHO-CCA, we also used early stopping on the validation set for best correlation. The GD hyperparameters appear in Table 5.

Table 5: OSHO-CCA's Hyper Parameters for Each Dataset

| Dataset | Epochs | Learning Rate | Weight Decay |
|---|---|---|---|
| Caltech | 5000 | $10^{-3}$ | 0 |
| NUS-WIDE | 4000 | $10^{-1}$ | 0 |
| Handwritten | 5000 | $10^{-3}$ | 0 |
| Cora | 1500 | $10^{-1}$ | $10^{-5}$ |

# E  Computation Details

We implemented our model in Pytorch and ran the experiments on a Linux server with NVIDIA A40 48GB GPUs and Intel(R) Core(TM) i7-8700 CPU 3.20GHz CPU.

