# OpenReview forum: "OSHO-CCA: Orthogonal and Scalable High-Order Canonical Correlation Analysis"
_TMLR — Rejected by TMLR_

### Review · Reviewer_akMK · 2026-01-04

**Summary Of Contributions:**

The paper considers the problem of multi-view CCA, where we compute a linear projection of multiple views to maximize joint correlation, while ensuring orthogonality of the projections. The current SOTA is TCCA, which is slow and does not strictly satisfy orthogonality. The paper proposes to optimise the TCCA using simple gradient descent while using a orthogonal projection parameterisation.

**Additional Comments:**

The hyperparameters of different datasets vary a lot. It's odd that learning rates go from 0.1 to 0.001 and weight decay is turned off for many cases. Can you discuss learning dynamics and sensitivity to hyperparameters? It would be useful to plot training curve over epochs. Have you considered adaptive training heuristics (eg. cosine annealing).

**Audience:**

Yes

**Audience Explanation:**

CCA is one of classic data analysis tools, and any improvements are certainly interesting for the community.

**Broader Impact Concerns:**

No issues

**Claims And Evidence:**

Yes

**Claims Explanation:**

The claims are that the new method is scalable, orthogonal, and captures high-order correlations. All three have been clearly demonstrated with convincing experiments or arguments.

**Requested Changes:**

See below

---

> ### Author Response · Authors · 2026-02-24
>
> We thank the reviewer for his constructive feedback and for recognizing that our claims regarding OSHO-CCA's scalability, orthogonality, and high-order correlation capabilities are supported by clear and convincing experiments. Below, we address the reviewer’s specific points:
>
> **Weight Decay**: During tuning, increasing weight decay yielded no improvement in performance on the validation set. It is important to note that OSHO-CCA is a linear approach and lacks the extreme expressivity of deep learning frameworks; consequently, extensive regularization via weight decay was not necessary to prevent overfitting.
>
> **Learning Rate and Stability**: The approach demonstrated low sensitivity to the learning rate. We observed that the learning curves were smooth and not "bumpy," indicating stable optimization. Because of this stability, we were often able to utilize a larger learning rate (e.g., $0.1$) simply to reduce total training time without sacrificing convergence quality.

---

### Review · Reviewer_6RJY · 2026-01-15

**Summary Of Contributions:**

In this paper, the authors propose a novel method called OSHO-CCA, which for the first time unifies the three major challenges in multi-view learning: high-order correlation modeling, orthogonality constraints, and computational scalability. By utilizing the QR decomposition of learnable parameters, the method transforms the orthogonally constrained optimization problem into an unconstrained one, enabling efficient training via gradient descent.

**Audience:**

Yes

**Audience Explanation:**

Canonical correlation analysis is an important direction in machine learning. The authors propose OSHO-CCA, and transform the orthogonally constrained optimization problem into an unconstrained one, enabling efficient training via gradient descent.

**Claims And Evidence:**

Yes

**Claims Explanation:**

1. The paper clearly identifies and tackles a well-known yet unresolved trade-off in multi-view CCA.
2. The synthetic experiment in Figure 3 is compelling and pedagogically effective, demonstrating why high-order modeling is essential when pairwise correlations are insufficient.
3. The experimental validation is comprehensive, showing consistent advantages in correlation maximization, downstream task performance, orthogonality preservation (Fig. 4), and scalability (Fig. 5).

**Requested Changes:**

1. I suggest adding a brief theoretical discussion to the Methods section (e.g., in Sections 4.1 or 4.2) to justify the choice of QR decomposition over other orthogonalization techniques and to clarify its functional role within the optimization framework.
2. Although the paper validates the overall performance of OSHO-CCA, it lacks ablation studies on its core components. It is recommended to supplement the experiments with the following to clarify the contributions of each part:(1) Remove the QR-based orthogonalization structure and optimize only the high-order objective using gradient descent;(2) Retain the QR orthogonalization but replace the high-order objective with a pairwise correlation sum.. These ablations would clarify how much each component contributes to the final results
3. The application of PCA to all views before training is mentioned only in the experimental section and not motivated in the methodology. Given that linear dimensionality reduction can alter or weaken higher-order interactions, the authors should discuss whether and how this preprocessing could affect the validity of the comparisons, especially for methods designed to capture high-order structures.

---

> ### Author Response · Authors · 2026-02-24
>
> We thank the reviewer for recognizing that OSHO-CCA successfully resolves the major trade-off in linear multi-view correlation learning. We appreciate that the reviewer found the synthetic experiment compelling and our experimental validation of correlation, orthogonality, and scalability to be comprehensive. We have addressed the reviewer’s points below:
>
> **Ablation Study (Section 5.9)**: As requested, we have conducted an ablation study, which has been added to the manuscript in Section 5.9. These experiments demonstrate the necessity of the orthogonalization,  without it, the model fails to produce the decorrelated embeddings essential for meaningful multiview representation.
>
>
> **Role of QR Decomposition**: We have added a brief clarification in Section 4.2 to clarify the functional role of the QR decomposition. We agree with the reviewer’s remark that there are indeed other orthogonalization methods that can fit in the framework. However, we view the QR primarily as a functional tool to perform orthogonalization in a differentiable way, allowing us to transform a constrained optimization problem into an unconstrained one. We believe the focus should remain on the overall optimization framework rather than the specific decomposition technique itself.
>
> **Motivation for PCA**: While the use of PCA is a practical step, we observed that it led to better TCC performance. Capturing high-order interactions is a significant challenge for a linear model. Our finding suggests that PCA serves as a regularization technique and makes the optimization challenge more manageable for the linear approach, allowing the model to more effectively align the views into a common subspace.

---

### Review · Reviewer_rpDf · 2026-02-12

**Summary Of Contributions:**

This paper proposes a new formulation for multiview canonical correlation analysis that focuses on capturing high-order correlations across more than two views, enforcing orthogonality among learned canonical components, and scalability with increasing number of views. The proposed method, OSHO-CCA avoids explicit tensor construction by optimizing a high-order correlation objective, while enforcing orthogonality based on QR decomposition. Furthermore, the authors introduce a new metric for evaluating multiview correlation.

The paper is positioned as an improvement over classical pairwise CCA variants and TCCA. The evaluation across several multiview datasets demonstrates that OSHO-CCA achieves competitive or improved downstream classification accuracy compared to selected baselines, maintains orthogonality as the number of views grows, and scales more efficiently than tensor-based methods.

Strengths

- The preliminaries and problem formulation are comprehensive, and the proposed method follows naturally from the limitations identified in prior CCA and TCCA approaches.
- The method is conceptually clean and technically well-motivated.
- Experimental results demonstrate clear advantages over TCCA in terms of orthogonality preservation and scalability.
- The paper includes qualitative visualizations that help illustrate correlation structure and orthogonality behavior across methods.

Weaknesses

- The method focuses exclusively on shared representations, while in realistic multiview settings, different views often contain both shared and private information. The absence of any explicit modeling or analysis of private components limits the applicability of the approach.
- The evaluation is limited in scope and primarily restricted to simple linear settings. Although the authors acknowledge this limitation, it remains unclear how OSHO-CCA would perform when applied to modern pretrained embeddings or more expressive representation spaces obtained from self-supervised training. Several existing self-supervised approaches also explores multiview/multimodal view with either geometric or information independence [1, 2, 3]
- The datasets used are relatively standard and limited, and all evaluations focus on classification tasks. In multiview settings, different views often complement each other for diverse objectives (e.g., localization, retrieval, or structured prediction), and evaluating on additional task types would strengthen the paper.
- While the synthetic experiments cover some ‘challenging’ cases (noise, pairwise independence), they don’t evaluate robustness to view corruption/missing views or distribution shift common in real-world multiview settings. The paper’s current evaluation seems to only focus on regular conditions and does not explore robustness under such conditions.
- Baselines are appropriate for linear multiview CCA, but the paper’s broader framing on multiview representations would benefit from discussion/connection to modern deep multiview representation learning.

**Minor Issues**

- Typo in the abstract: OSHOCCA → OSHO-CCA
- Section 4.3: metricwe → metric we
- Section 5.3: beslines → baselines

[1] Tian, Yonglong, Dilip Krishnan, and Phillip Isola. "Contrastive multiview coding." In *European conference on computer vision*, pp. 776-794. Cham: Springer International Publishing, 2020.

[2] Jiang, Qian, Changyou Chen, Han Zhao, Liqun Chen, Qing Ping, Son Dinh Tran, Yi Xu, Belinda Zeng, and Trishul Chilimbi. "Understanding and constructing latent modality structures in multi-modal representation learning." In *Proceedings of the IEEE/CVF Conference on Computer Vision and Pattern Recognition*, pp. 7661-7671. 2023.

[3] Kimura, Tomoyoshi, Xinlin Li, Osama Hanna, Yatong Chen, Yizhuo Chen, Denizhan Kara, Tianshi Wang et al. "Infomae: Pair-efficient cross-modal alignment for multimodal time-series sensing signals." In *Proceedings of the ACM on Web Conference 2025*, pp. 3084-3095. 2025.

[4] Liang, Paul Pu, Zihao Deng, Martin Q. Ma, James Y. Zou, Louis-Philippe Morency, and Ruslan Salakhutdinov. "Factorized contrastive learning: Going beyond multi-view redundancy." *Advances in Neural Information Processing Systems* 36 (2023): 32971-32998.

**Audience:**

Yes

**Audience Explanation:**

The paper studies multi-view correlation, which could be interesting, but only to a limited set of audience for TMLR.

**Broader Impact Concerns:**

No ethical or broader impact concerns were identified.

**Claims And Evidence:**

No

**Claims Explanation:**

While the proposed method demonstrates improvements over traditional CCA and TCCA variants with respect to orthogonality, scalability, and high-order correlation modeling, the broader claims made in the abstract and introduction, particularly regarding learning multiview representations, are not fully supported by the current evaluation. The evidence supports the claims within the linear multiview CCA setting, but not the broader representation learning. Given the state of the art in multiview and multimodal representation learning, which increasingly relies on deep and nonlinear models, the empirical evidence provided is weakened and does not fully justify the broader representational claims.

**Requested Changes:**

Most of the requested changes match the weakness above. Below are some recommended changes suggested from most critical to least critical:

1. Broaden the empirical evaluation beyond linear classification, ideally including experiments on modern pretrained embeddings or more expressive representation spaces.
2. Include robustness analyses, such as experiments with view corruption, noise, or mismatched correlations, to evaluate robustness in realistic multiview settings.
3. Discuss more recent multiview or representation learning baselines, particularly deep or hybrid approaches.
4. Clarify the scope of claims in the abstract and introduction to better align with the linear and correlation-focused nature of the current evaluation.
5. Explore or discuss shared–private decomposition, which is highly relevant in multiview learning contexts.

---

> ### Author Response · Authors · 2026-02-24
>
> We thank the reviewer for his thoughtful feedback and for recognizing that OSHO-CCA is conceptually clean, technically well-motivated, and follows naturally from the limitations of prior approaches. We are particularly pleased that the reviewer found our problem formulation comprehensive and our experimental results and qualitative visualizations clearly demonstrate the method's advantages in orthogonality preservation and scalability. We have addressed the reviewer’s points below:
>
> **Pretrained embeddings**: To address the reviewer’s comment regarding the performance on modern pretrained embeddings, we have introduced a new variant of the Cora dataset, denoted as Cora_SBERT. In this variant, we replaced the original sparse bag-of-words text representation with dense embeddings generated by Sentence-BERT (SBERT). As demonstrated in our updated evaluations, OSHO-CCA still successfully attains the highest TCC across all baselines, demonstrating a particularly strong advantage over TCCA.
>
> **Scope of the claims and shared-private methods**: We thank the reviewer for highlighting this important distinction. We agree that the broader terminology used in the original manuscript could lead readers to expect a general-purpose or deep multiview representation learning framework.
>
> Our primary intention is to stay strictly focused on the domain of linear multiview correlation learning. We observed specific, fundamental limitations in the current linear multiview CCA literature, namely, the inability of existing methods to simultaneously achieve scalability and orthogonality while capturing high-order interactions. Our goal with OSHO-CCA was to tackle these precise mathematical bottlenecks. By solving these foundational issues, we aim to establish a mathematically sound baseline that opens up a new direction of research, upon which future non-linear/deep learning or shared-private extensions can confidently be built.
> To accurately reflect this, we have revised the Abstract and Introduction to explicitly frame our contributions around "linear multiview correlation" and have clarified that our evaluations focus specifically on linear downstream tasks and correlation maximization.
>
> **Robustness evaluation**: We appreciate the reviewer's perspective on robustness. In total, we have conducted six different experimental setups to thoroughly evaluate the properties and downstream performance of OSHO-CCA. While evaluating robustness against view corruption, missing views, or distribution shifts is an important topic for realistic multiview settings, we consider this a distinct and separate research direction. Our current scope is deliberately focused on solving the foundational challenges of high-order correlation and scalability under regular conditions.
>
> **Deep frameworks vs. linear focus**: We acknowledge the importance and rapid progress of modern deep multiview representation learning. However, our proposed approach explicitly aims to address the theoretical and computational bottlenecks, namely the lack of orthogonality and limited scalability, found in classical linear high-order methods. By maintaining a strictly linear framework, we provide a clean, mathematically principled foundation to solve these inherent issues. As noted in our conclusion, extending OSHO-CCA to a nonlinear variant using deep neural networks is a highly promising direction for our future work, but establishing a linear baseline was a necessary prerequisite.

---

### Review · Reviewer_zdAV · 2026-02-24

**Summary Of Contributions:**

This paper proposes OSHO-CCA, a multiview CCA method that claims to simultaneously achieve three desiderata: high-order correlation, scalability and orthogonality. The method combines TCCA's high-order correlation objective (element-wise product form) with QR decomposition for orthogonality enforcement, optimized via gradient descent in PyTorch. A new Total Canonical Correlation (TCC) evaluation metric is also introduced. Experiments on four benchmark datasets and synthetic data compare against SUMCOR-GCCA, MAXVAR-GCCA and TCCA.

**Audience:**

Yes

**Audience Explanation:**

Multiview CCA is a relevant topic with applications in neuroscience, genomics and multimodal learning. The problem framing would be of interest to practitioners working with multi-view data.

**Claims And Evidence:**

No

**Claims Explanation:**

The paper makes three central claims: (1) being the first method to simultaneously achieve high-order correlation, scalability, and orthogonality is a meaningful contribution, (2) OSHO-CCA addresses a real gap in multiview CCA, and (3) it outperforms existing methods. The evidence falls short on all three.
1. The paper positions itself as the first to achieve all three properties (Fig 1), but provides no analysis of why this combination was not previously realized. The proposed method combines TCCA's element-wise product objective with QR parameterization, i.e., two fully compatible and previously known components. The absence of prior work combining them appears to reflect a gap in the literature rather than a genuine technical barrier, which may limit the significance of being the first method to realize it.
2. Orthogonality, the paper's distinguishing design choice, lacks empirical justification for its practical importance. The theoretical necessity of orthogonality to avoid degenerate solutions is well-established and looks trivially correct in the CCA/PCA literature without a requirement of argumentation. What the paper actually needs to demonstrate may be that TCCA's lack of orthogonality causes meaningful degradation in practice. The existing results seem suggesting otherwise: TCCA remains competitive on several datasets (e.g., 0.930 vs 0.944 on Handwritten), and Fig 4 shows its covariance matrices remain near-diagonal for 2-3 views, with deviation only emerging gradually as views increase. A critical ablation of OSHO-CCA with vs without QR constraints is missing and would be necessary to isolate orthogonality's independent contribution to performance.
3. The proposed TCC metric possibly has circular bias. TCC applies Gram-Schmidt orthonormalization before computing correlation, inherently favoring methods that enforce orthogonality. OSHO-CCA's dominance on this metric may be therefore expected. Raw high-order correlation without Gram-Schmidt is not reported, making it hard to assess whether the improvement reflects genuine alignment or the built-in evaluation bias.
4. Classification accuracy results may not support the "outperforms" claim. OSHO-CCA loses to SUMCOR-GCCA on NUS-WIDE and Cora, and wins on Caltech and Handwritten within overlapping standard deviations. No statistical significance tests are conducted despite 5 runs. Moreover, Cora's variance (\pm 0.053) makes the conclusion unreliable. This profile may only support "comparable".
5. No ablations disentangle the three claimed contributions. High-order modeling, orthogonality and the optimization strategy are not independently validated. Performance gains cannot be attributed to any specific component, making it hard to understand why or when OSHO-CCA helps.
6. Scalability evidence may be insufficient. Experiments reach only 6 views with dimension 20. For a method positioning scalability as a core contribution, this scale is not convincing enough to demonstrate practical advantage.

**Requested Changes:**

Please consider addressing all my concerns in the above section, especially:
1. Add ablation: compare OSHO-CCA with vs without QR orthogonality constraints to isolate the contribution of each design choice.
2. Report raw high-order correlation (Eq 2) without Gram-Schmidt alongside TCC to provide a neutral evaluation.
3. Conduct statistical significance tests for all comparisons given the small number of runs.
4. Scale the scalability experiment to substantially more views and higher dimensions.
5. Reduce the claim "outperforms" in the abstract to match the actual experimental profile.

---

### Author Response · Authors · 2025-12-17
**Review Process Status**

We have noticed that the review process has not started yet. We kindly ask the AE whether he finds the manuscript appropriate for review, so the review process can start.

---

> ### Comment · Action_Editor_g4cA · 2025-12-17
> **Approved for review**
>
> Dear authors, your manuscript has been approved for review. Recruitment of reviewers will begin soon.

---

### Author Response · Authors · 2026-02-24

We sincerely thank the reviewers for their constructive feedback and insightful suggestions, which have greatly improved the quality of our work. We have addressed each comment point-by-point below and highlighted all modifications and new information in red within the revised manuscript.

---

### Author Response · Authors · 2026-03-16

Dear Jaakko,
We'd be happy to kindly inquire about the status of our article, as we did not get any communication from TMLR after submitting our author responses on Feb 24.

Thank you

---

> ### Comment · Action_Editor_g4cA · 2026-03-16
> **A recommendation will be submitted this week**
>
> Thank you for reaching out. A recommendation based on the available reviews and comments will be submitted this week which will then move the process forward.

---

> > ### Author Response · Authors · 2026-03-31
> >
> > Dear Jaako,
> > We would like to kindly ask again about our article's status,
> >
> > Thank you

---

> > > ### Comment · Action_Editor_g4cA · 2026-03-31
> > > **A recommendation will be posted this week**
> > >
> > > Dear authors, a recommendation on the article will be posted during this week.

---

### Decision · Action_Editor_g4cA · 2026-05-03

**Recommendation:** Reject

**Additional Comments:**

The paper received mixed reviews, with one reviewer leaning towards rejection, one leaning towards acceptance, and a third recommending to accept. (While four reviews were received, recommendations were received for three of the reviews.)

The reviewers appreciated several aspects of the work.

+ CCA overall was considered an important direction [6RJY], the multiview CCA topic was considered relevant [zdAV], and the addressed tradeoff was considered clearly identified [6RJY]
+ The problem formulation was considered comprehensive [rpDf]
+ The method was seen as following naturally from limitations of CCA and TCCA [rpDf], and was seen as well motivated and conceptually clean [rpDf]
+ The synthetic experiment was appreciated [6RJY]
+ The experimental validation was considered comprehensive by one reviewer [6RJY] and the experiments were considered to convincingly demonstrate the claims [akMK]
+ Qualitative visualizations were appreciated [rpDf]


However, several concerns were raised; the authors provided rebuttals for many of the concerns.

- Further commentary on novelty was desired, to address why the combination of three properties was not realized in previous work [zdAV]
- Focusing on shared representations only and not modeling private components was criticized [rpDf]; authors commented on their intenrion to focus on linear  methods (unclear how that addresses the shared and provate components concern).
- Discussion of QR decomposition versus other orthogonalization techniques was desired [6RJY]; authors added a brief clarification.
- Discussion of relationship to deep multiview representation learning was desired [rpDf]; authors commented on their intenrion to focus on linear  methods.
- Lacking motivation for the importance of orthogonality was criticized [zdAV]
- An ablation without QR constraints was desired to isolate contribution of orthogonality [zdAV,6RJY]; authors added an ablation study.
- Ablations for other contributions were also desired [zdAV,6RJY]
- Discussion of the impact of PCA preprocessing was desired [6RJY]; authors provided a brief response.
- The evaluation was considered limited in scope [rpDf] and potential performance in modern embedding or other expressive representation spaces was considered unclear [rpDf]; authors provided some updated evaluations with a modified dataset with dense embeddings.
- The evaluation data sets were considered limited [rpDf] and use of classification tasks only was criticized [rpDf]
- Evaluation of robustness to conditions such as view corruption, missingness and distributions shift were desired [rpDf]; authors considered it a separate research direction.
- Discussion of learning dynamics and sensitivity to hyperparameters was desired [akMK]; authors provided some comments on weight decay, stability and learning rate.
- A concern was raised that the TCC metric could have circular bias favoring methods with orthogonality [zdAV]
- The results were criticized as not clearly showing OSHO-CCA to outperform others [zdAV]
- Lack of statistical significance tests was criticized [zdAV]
- Evidence of scalability was criticized as insufficient [zdAV]

After the author response period, concerns still remained. Concerns of [zdAV] were considered mostly to be unchanged, with one provided ablation considered a partial response only, and concerns of lacking evidence were considered to still remain [zdAV].
Similarly, despite the author response [rpDf] still considered the empirical evaluation narrow and robustness to be unclear, and relation to modern multiview literature to be superficial only. On the other hand, the contributions to CCA were considered clear and empirically backed [akMK].

**Audience:**

Yes

**Audience Explanation:**

CCA-style settings remain relevant as noted by [akMK]. It seems clear that the paper could be of interest to at least part of the TMLR audience, particularly practitioners working with multiview data even if that is a limited part of the audience, as noted by [zdAV,rpDf]. The reviewers also noted several individual positive aspects of the paper (see Additional Comments) which could point to interest in part of the TMLR audience.

**Claims And Evidence:**

No

**Claims Explanation:**

Here, the reviewers' opinion is mixed, and my judgement here is essentially that the paper remains in an unclear state regarding sufficiency of the evidence for the claims.

In particular, one reviewer still felt after the current responses and changes of the authors that the empirical evaluation was narrow and robustness not fully clear [rpDf], and also positioning with modern multiview methods is not thorough enough [rpDf]. Several issues of experimental rigor noted by another reviewer [zdAV] were also not yet addressed; while the authors correctly noted that this latest review was submitted later than the others, and stated their response was aimed at the first three reviews, I must still acknowledge the raised concerns in the fourth review too.

While the authors' edits have narrowed the scope of the claims, and TMLR's criteria are about whether the claims made are supported by the evidence, essentially based on the current reviews concern remains that even the narrowed claims might not be sufficiently supported; in particular, issues of robustness and rigor (e.g. potential bias in an evaluation metric; statistical significance reporting; scalability exploration).

Overall, I believe with small further additions and a re-evaluation by reviewers the paper could be made clearly acceptable in terms of the evidence, but in its current state it is not quite there yet. This is somewhere between a major and minor revision, but I think it would benefit from re-reading by reviewers and not only checking by an editor. Thus, I am currently recommending rejection with strong suggestion to submit a revised version.

**Resubmission Of Major Revision:**

The authors may consider submitting a major revision at a later time.